# Potential Use of Tropical and Subtropical Fruits By-Products in Pig Diet: In Vitro Two-Step Evaluation

**DOI:** 10.3390/ani15101454

**Published:** 2025-05-17

**Authors:** Dieu donné Kiatti, Francesco Serrapica, Nadia Musco, Rossella Di Palo, Serena Calabrò

**Affiliations:** 1Department of Veterinary Medicine and Animal Production, University of Napoli Federico II, Via F. Delpino 1, 80137 Naples, Italy; dieudonne.kiatti@unina.it (D.d.K.); dipalo@unina.it (R.D.P.); serena.calabro@unina.it (S.C.); 2Department of Agricultural Sciences, University of Naples Federico II, Portici, 80138 Naples, Italy; francesco.serrapica@unina.it

**Keywords:** mango, cashew, pineapple, gas production, enzymatic digestion, monogastric nutrition

## Abstract

Pineapple, cashew and mango are among the most cultivated plants in tropical and subtropical regions due to the high demand around the world. The harvesting and processing of the fruits of these plants generate a huge amount of waste. This study aims to propose alternative feeds for pigs by characterizing cashew, pineapple and mango fruit by-products through an in vitro two-step (gastro-intestinal and caecum) study to provide feeds not competing with humans and promoting eco-sustainable livestock.

## 1. Introduction

According to the Food and Agriculture Organization of the United Nations, approximately 1.3 billion tons of food are wasted worldwide every year from harvesting to processing, bypassing fresh consumption. Fruits, vegetables, roots and tubers account for approximately 80% [1,2] of total yearly waste due to their perishability. In tropical and subtropical regions, pineapple, cashew and mango cultivations contribute significantly to the waste generation [3,4].

Pineapple (*Ananas comosus* L.) is native to tropical and subtropical areas (Asia, South and North America, Africa and Oceania), with Cayenne, Sugarloaf, Spanish, Queen, Pernambuco and Perolera the main cultivated varieties [5,6]. In 2021, global pineapple production was estimated at 28,179,348 tons [4]. Costa Rica, the Philippines and Brazil are the main contributors, whereas in West Africa, Nigeria and Benin are ranked 8th and 17th, with 1,671,440 and 386,906 tons produced, respectively [7]. Considering the amount of pineapple globally produced, more than 18 million tons of pineapple waste are generated each year, generally, because of the short shelf-life, high moisture and sugar content.

The cashew tree (*Anacardium occidentale* L.), native to tropical South America, has also spread to other tropical and subtropical areas (Asia, Africa and Central America) due to the high demand for its products, mainly nuts [7]. It includes about 60 genera and 400 species and two principal varieties (red and yellow) in accordance with the peduncle (apple or false fruit) color. The nut is the real fruit and does not present a difference between varieties and represents the interest product of cashew plantations [4]. The nut represents 10% of the total fruit, while the apple represents 90%, and it is less processed due to its perishability and astringency taste [8]. The world cashew nut production was estimated at 5,535,510 tones with approximately one-third coming from West African regions [9]. The cashew apples processed generate the pomace, whereas the nuts generate shell and testa [10].

Mango (*Mangifera indica* L.) is one of the most important tropical fruits in the world and currently ranked fifth in total world production among the major fruit crops due to its succulence, exotic flavor, delicious taste and the nutrients it contains [11,12]. Mango production is continuously increasing (from 35.7 × 106 tons in 2009 to 55.9 × 106 tons in 2019) with African countries accounting for about 40%, mainly supported by the West African region [13,14]. Mangos are consumed freshly (80%) or processed (20%) into puree, nectar, leather, pickles, canned slices and chutney, generating 50–65% of fruits as waste composed of seeds (endocarp: kernel + testa + shell) and peels [12,15].

In general, fruit by-products, such as from cashew, pineapple and mango, are thrown into the environment contributing to global warming, water pollution and, as a consequence, potential human health problems. Such problems could be reduced by recycling these wastes in other fields, thus creating a circular economy. In West African conditions, one of the promising fields should be the animal production sector by facing the challenge to feed animals according to their requirements due to the high feeding costs (60–85%) and cereals competition [3,16]. The selection of the most appropriate raw materials and the formulation of a feeding plan are two factors that can influence the efficiency indicators in livestock farms [17]. Strategies and solutions, such as a “food recovery hierarchy”, are therefore needed to reduce the environmental impact of feed production and the use of natural resources by increasing their reuse [18]. Moreover, there are many factors limiting the use of unconventional feed, such as limited knowledge and high variability of their chemical composition, high polyphenols and moisture content [19,20]. Previous studies, even limited, reported that cashew, pineapple and mango by-products are useful as alternative feedstuff in West African livestock for small ruminants [3,4], growing and finishing pigs [21], laying hens [22] and dairy cows [23]. Indeed, there is still a need to provide more knowledge to these by-products for their wide adoption in animal nutrition. In this context, the aim of the present research was to contribute to the characterization of tropical and subtropical fruit by-products (cashew, pineapple and mango) for pig nutrition through an in vitro two-step (gastro-intestinal and caecum) study to provide feeds not competing with humans and promoting eco-sustainable livestock.

## 2. Materials and Methods

### 2.1. Materials and Locations of the Study

The study was performed in 2024 collecting No. 10 by-products derived from the processing of pineapple (No. 2), cashew (No. 5) and mango (No. 3). Different areas of Benin were chosen for sampling, based on their great suitability for growing the fruits. All the samples were oven-dried at 65 °C for 24 h and milled with a 1.1 mm screen for chemical composition analysis and in vitro two-step evaluation using caecal content from growing pigs. In addition, three feeds commonly used in the pig diet [corn meal (CM), soybean meal (SBM) and beet pulp (BP)] were also selected as reference feeds for their already known nutritional suitability in pigs.

### 2.2. Samples Collection

Pineapple samples (var. Smooth Cayenne) were collected in the Atlantic district, located in the sub-humid zone of Benin (6°18′–6°58′ N and 1°56′–2°30′ E) and characterized by two rainy seasons and two dry seasons (average annual rainfall: 1200 mm; temperature ranging from 27 to 31 °C; relative humidity ranging from 69 to 97%) [24]. A total of 40 fruits were collected randomly from three different farms. Following local traditions, the fruits were cleaned by removing the crowns and buds using a steel knife. Then, the peel and core were delicately separated from the pulp, which was pressed using a hydraulic press to remove the juice and retain the pomace. Two samples were selected from the pineapple waste generated: pineapple peel (P_Peel) and pomace (P_Pom).

Cashew apple samples (var. yellow and red) were collected in the municipality of Parakou in the north-east of Benin (8.55′–10.53′ N and 2°–3°50′ E), in the Sudanian Zone, characterized by one dry and one rainy season (annual rainfall ranging from 900 to 1000 mm; mean relative temperature and humidity 27.5 °C and 54.9%, respectively) [25]. Cashew apples were directly collected after nut harvesting and separating the varieties (red and yellow). The apples were processed into juice using a hydraulic press machine (Tianyu Youdo Machine, model UDZL-W33, dimensions 330_330_600 mm, Kaifeng, China), generating pomace as waste. In addition to the pomace, the whole apples per variety were also collected due to their availability, as well as at the cashew plantation as nuts, harvesting the waste. So, four samples were collected from cashew apples: cashew yellow whole fruit (CY_WF), cashew yellow pomace (CY_Pom), cashew red whole fruit (CR_WF) and cashew red pomace (CR_Pom). In addition, another nut processing waste, cashew nut testa (CNT), was sampled at the factory and added to the cashew apple sample for the study.

Mango samples were collected in the municipality of Natitingou in the north-west of Benin (10°18′ N and 1°22′ E), in the Sudanian Zone characterized by one dry and one rainy season (annual rainfall ranging from 900 to 1000 mm) [26]. About 50 kg of fresh fruit was collected from five different mango plantations and processed by peeling and removing the endocarp. The waste generated from mango fruit processing was peel, kernel, taste and husk. The mango peel (M_Peel), kernel (M_Ker) and testa (M_T) were sampled for the study.

### 2.3. Chemical Composition

The samples were analyzed according to the official procedures of the Association of Chemists [26] to determine dry matter (DM, ID 934.01), the composition of ether extract (EE, ID 920.39), crude protein (CP, ID 2001.11) and ash (ID 942.05). The structural carbohydrate fractions, such as neutral detergent fiber (NDF), acid detergent fiber (ADF) and acid detergent lignin (ADL) were also determined [27] and expressed excluding the residual ash.

### 2.4. Experimental Design and In Vitro Two-Step Digestibility Evaluation

To determine the digestibility in monogastric animals, a two-step in vitro method was used, where the first phase (Step 1) simulates the digestive processes occurring in the stomach and small intestine by the action of pepsin and pancreatin enzymes [28], and the second phase (Step 2) reproduces the intestinal microbial activity happening in the large intestine by symbiotic bacteria [29].

Step 1: Approximately 2 ± 0.0024 g of each substrate was weighed in 4 replicates into 300 mL flasks; a blank was included in each series. A total of 50 mL of phosphate buffer (0.1 M, pH 6.0) and 20 mL of HCl (0.2 M) were added to each flask, keeping the flasks under continuous agitation, after having adjusted the pH of each flask to 2.0 using 1 M HCl and 1 M NaOH. Finally, 2.0 mL of a freshly prepared solution containing 10 mg of pepsin (Sigma, Tokyo, Japan, P7000-100g) was added. The flasks were closed with rubber stoppers and then incubated under continuous agitation for 2 h in a thermostatic bath at 39 °C (Incubator SI60; Stuart Scientific, Stone, Staffordshire, UK). At the end of the incubation, the flasks were cooled, and 20 mL of phosphate buffer (0.2 m, pH 6.8) and 10 mL of 0.6 M NaOH buffer solution were added. The pH was again adjusted at 6.8 with 1 M HCl and 1 M NaOH; then, 2 mL of a freshly prepared pancreatin solution containing 200 mg of powdered pancreatin (Sigma, P1225-100g) was added to each flask. The flask contents were mixed and placed in a thermostatic bath to be incubated at 39 °C for 4 h with continuous shaking. The undigested residue was collected in pre-weighed Whatman 934-AH filters. The filters containing the undigested substrate were dried at 65 °C for 24 h and weighed. The substrates were collected for the in vitro fermentation (Step 2).

Step 2: The residual material obtained in Step 1 was incubated with porcine caecal content. For this purpose, approximately 0.5 ± 0.0026 g of Step 1 predigested material of each sample was weighed in three replications into 120 mL serum bottles. Samples of caecum content were collected during slaughter in an EU-authorized slaughterhouse [30], from four 12-month-old, 100 kg fattening pigs (Landrace × Large White crosses) fed commercial feed (CP: 14.8%; gross energy: 15.0 MJ/kg). The material collected in heated thermoses (at 39 °C) was rapidly transported to the laboratory, where it was pooled, filtered through a double layer of gauze, diluted (1:6) in NaCl solution and added to each bottle (5.0 mL) already containing the medium (79 mL) under a continuous flow of CO_2_ prepared according to Boisen et al. [28]. The bottles were incubated at 39 °C into a thermostat under anaerobic conditions, and, depending on the chemical composition (particularly concerning the structural and non-structural carbohydrates) and also observing the fermentation trend during the trial, the incubation was stopped at different times: (i) at 48 h: CNT, CY_WF CY_Pom, CR_WF, CR_Pom and CM; and (ii) at 72 h: P_Peel, P_Pom, M_Peel, M_Ker, M_T, SBM and BP. During the incubation, the gas pressure and volume were manually measured at 3–24 h intervals using a pressure transducer (Cole and Parmer Instrument Co., Vernon Hills, IL, USA). Dry matter digestibility (DMD) was determined by filtering the contents of the bottle through glass crucibles (Scott Duran porosity #2) and drying them for 24 h at 103 °C. The fermentation liquid was collected from each bottle to determine the pH using a pH-meter (model 3030 Alessandrini Instrument glass electrode, Jenway, Dunmow, UK) and the production of short-chain fatty acids (SCFAs: acetic, propionic, butyric, iso-butyric, iso-valeric, and valeric) using gas chromatograph (Thermo Fisher Scientific, Rodano, MI, Italy; model 1310 trace).

### 2.5. Data Processing and Statistical Analysis

Non-structural carbohydrates (NSCs, % DM, 103 °C) were calculated according to Vas Soest et al. [27], as follows:NSC = DM − NDF − CP − EE − ash(1)

Total dry matter digestibility (DMDt, %) was calculated based on the data obtained in Step 1 (DMDe, enzymatic digestibility, %) and Step 2 (DMDm, microbial digestibility, %) according to Boisen et al. [28], as follows:DMDt = [(100 − DMDe) × (DMDm/100)] + (DMDe)(2)

Gas and short-chain fatty acid production (at 48 and 72 h) were plotted against grams of incubated organic matter (OMCV, mL/g and SCFA, mmol/g, respectively). The proportion of branched-chain fatty acids (BCFAs) was calculated according to Musco et al. [29], as follows:BCFA = [(iso-valeric + iso-butyric)/SCFA](3)

The software JMP^®^ (Version 14 SW, SAS Institute Inc., Cary, NC, USA, 1989–2019) was utilized to analyze the data. Digestibility data and in vitro fermentation data for the two incubation times were analyzed separately using Student’s *t*-test to test the effect of the feedstuff as a single factor, according to the model:yij = μ + Fi + εij(4)
where y is the tested parameters, μ the general mean, F feedstuffs factor (i = 1–10) and ε is the error term.

## 3. Results

### 3.1. Chemical Composition

The chemical composition of the by-products is reported in Table 1. The average dry matter content—excluding CNT, CM and SBM (91.4, 86.7 and 89.3%, respectively)—was low (24.8 ± 13.7%). The ash content among the African by-products was slightly high only in P_Peel (5.95% DM), even if it was lower than SBM (7.77% DM). Regarding the fractions of structural carbohydrates, a group of by-products with a high content of cell wall (NDF higher than 25% DM) was identifiable (P_Peel, CY_Pom, CR_Pom, M_T, BP), where the lignin content was highly variable (ADL range: from 1.83 to 11.56% DM). In all the other feedstuffs, NDF values fall between 20.32 and 7.01% DM, but the ADL content was in some cases rather high; as expected, only CM appeared particularly poor in all fiber components (NDF, ADF, ADL: 7.58, 2.83, 2.40% DM, respectively). On the other hand, the average values of non-structural carbohydrates (NSCs), excluding SBM (14.6% DM), were higher (54.3 ± 13.1% DM). Regarding lipids, only CNT, followed by M_Ker, showed a high content (EE: 26.4 and 11.2% DM, respectively), and excluding the above-mentioned, they varied between 0.27 and 4.73% DM. The protein content, as known, was very high in SBM (51.2% DM), fair in pineapple and cashew by-products (average value: 8.55 ± 1.60% DM) but very low in mango by-products (M_Peel, M_Ker and M_T: 2.51, 4.69 and 3.97% DM, respectively).

### 3.2. In Vitro Digestibility

Table 2 shows the percentage values of in vitro dry matter digestibility, reported as enzymatic, microbial and total. For both incubation times considered, 48 and 72 h, the values of the three parameters differ significantly (*p* < 0.001) between substrates and, as expected, they are overall highest in the longer incubation time. Among those stopped at 48 h, the total digestibility was significantly higher in corn meal and lower in cashew testa (88.3 and 28.5%, CM and CNT, respectively; *p* < 0.001); the same trend was observed in enzymatic and microbial digestibility. It was also observed that, in some by-products (CY_WF, CR_WF, CR_Pom), the enzymatic digestibility was more consistently higher than in the microbial one, while in others (CNT, CY_Pom, CM) the difference was minimal. Among the samples stopped at 72 h, the total digestibility was particularly high in SBM (94.2%; *p* < 0.001) and significantly lower in M_T (33.6%, *p* < 0.001); the same trend was observed in enzymatic and microbial digestibility. In addition, in some feeds (P_Peel, P_Pom), the enzymatic digestibility was similar to the microbial one, while in most of these (M_Peel, M_Ker, M_T, SBM and BP) the enzymatic digestibility was lower than the microbial one.

### 3.3. In Vitro Fermentation Characteristics

Table 3 shows the in vitro fermentation characteristics obtained after 48 and 72 h of incubation. At both fermentation times, the values of all the parameters considered are highly and significantly different (*p* < 0.001) between the substrates and overall higher in the longer incubation. At 48 h, the CM (OMCV: 142 mL/g; *p* < 0.001) was the substrate that produced the highest gas, and CNT (OMCV: 27.3 mL/g; *p* < 0.001) was the one that produced the least. The same trend was observed for SCFA where acetate and propionate were the most represented. Butyric acid was highest in CM (26.0 mmol/g, *p* < 0.001). The branched-chain fatty acids (BCFAs) were significantly higher (*p* < 0.001) in cashew apple by-products (10.6, 9.66 and 11.4, 6.84, for CY_WF, CY_Pom, CR_WF and CR_Pom, respectively). After 72 h of fermentation, the substrates that produced the highest gas were P_Pom, M_Ker and BP (*p* < 0.001), and the one that produced the least was M_T (*p* < 0.001). The trend was similar for SCFA even if no statistical significance emerged. Butyric acid production was significantly higher in M_Ker (25.7 mmol/g; *p* < 0.001). The proportion of BCFA was significantly higher in M_T and lower in P_Pom (6.23 and 2.67; *p* < 0.001, respectively).

### 3.4. In Vitro Fermentation Kinetics

In Figure 1 and Figure 2, the gas production profiles of the studied samples incubated for 48 and 72 h are, respectively, represented. The by-products incubated for a shorter time, 48 h, [cashew nut testa, cashew (var. yellow) whole fruit, cashew (var. yellow) pomace, cashew (var. red) whole fruit; cashew (var. red) pomace; corn meal] showed similar and lower curves than corn meal incubated as a reference feed (Figure 1), indicating that, compared to the most used cereal in pig diets, the fruit by-products incubated are poor of residual nutrients. For the by-products’ gas profiles, it was also possible to note that the curves were characterized by a biphasic shape, indicating the presence of nutrients that were fermented in two different phases: a faster one, which was exhausted at approximately 24 h, and a slower one, which probably goes beyond 48 h. The by-products incubated for a longer period (72 h) (mango peel, kernel and testa, pineapple peel and pomace), showed intermediate curves compared with soybean meal and beet pulp ones, incubated as reference feeds (Figure 2), and could be classified into three groups depending on the fermentation speed: (i) slow fermentation process with low gas production (M_T); (ii) medium fermentation rate with high gas production (M_Peel, P_Peels, P_Pom, SBM and BP); and (iii) rapid fermentation speed after the first 12 h of incubation and high gas production (M_Ker). Even at 72 h, it was possible to note that all the curves, except that of the M_Ker, showed a biphasic shape due to the presence of both fractions with fast and slow fermentability, even if in this case the timing is less clear.

## 4. Discussion

Discussing the nutritional characteristics, as well as the in vitro digestion and fermentation characteristics, of by-products derived from the three selected tropical and subtropical fruits (pineapple, cashew and mango) as promising ingredients in pig feeding plan is not easy due to the scarcity of data in the literature on similar studies. Furthermore, one of the major difficulties in comparing the data obtained with those of other authors lies in the great variation in processing methods to which they are subjected, often artisanal and linked to local processing methods. The pH values (ranging from 5.60 up to 6.32) for all tested substrates fall in the interval considered physiological in pig cecum-colon (5.2–7.8) [31]. This indicates that the in vitro buffer system had guaranteed an environment suitable for microbial fermentation. The referenced feeds, such as corn meal (CM), soybean meal (SBM) and beet pulp (BP), showed the variation in terms of the chemical composition of feedstuff that pigs can accept. The investigated by-products fall within the interval of referenced feeds for NDF (7.6–47.1% DM) and CP (2.5–51.2% DM), except the by-products from mango fruits that showed a low CP. However, only M_Peel, M_Ker and pineapple by-products (P_Peel and P_Pom) have a low ADL comparable to pig-referenced feeds. The negative effect of high fibers and their lignification on pig nutrient utilization and fermentation profile were previously reported [32,33]. This effect justifies the high differences observed for total digestibility and fermentation kinetics of the investigated by-products compared to CM, which were both stopped at 48 h of incubation. In addition, the by-products contain more nonstructural carbohydrates (NSCs) than SBM, and BP was not so far from the one of CM. A large portion of NSC is digested by the microflora of the large intestine in pigs, contributing up to 50% of the dietary energy [34]. This result was observed for the by-products stopped at 72 h of incubation, showing greater DMDm, SCFA and fermentation process as well.

### 4.1. Use of Pineapple By-Products in Pig Nutrition

Previous studies [3] indicated that the chemical composition of pineapple by-products is partially in accordance with other authors’ publications, and this is probably because the production areas and the soil-climatic conditions, as well as the type of processing, are different. In the literature, studies that used pineapple by-products in the formulation of diets for pigs are few, very dated and report unfavorable effects, especially in terms of low palatability and high fiber content [35]. The same author reported the improvement in body weight gain and feed conversion efficiency in older pigs when pineapple by-products were added to up to 50% in the diet but, beyond this level, these parameters were depressed. Wadhwa and Bakshi [36] noted that the pigs did not accept dried pineapple waste presented ad libitum and, due to the high crude fiber content (20%), limits its use in young pigs’ diets. Moreover, the decreases in the pigs’ average daily gain (17–24%) and feed intake (12–21%) were observed with pineapple by-products inclusion of up to 270 g/kg DM in the basal diet that was composed of corn and soybean meal, while feed conversion ratio was not affected [33]. The authors reported also an increase in insoluble and total dietary fiber digestibility and lean meat percentage, whereas the carcass fat deposition decreased with pineapple by-products inclusion in the pigs’ diets. Indeed, in this study, both pineapple by-products, specifically pomace, seemed to be promising. In particular, in pomace, the nutrient content and in vitro fermentation characteristics were similar to beet pulp in terms of protein content, structural carbohydrates, digestibility, gas and fatty acids production. So, this by-product could be a suitable ingredient for pigs’ feeding plan. Importantly, the low dry matter content is a critical issue to consider because preservation over time may be difficult.

### 4.2. Use of Cashew By-Products in Pig Nutrition

A slight difference between yellow and red varieties of cashew apple by-products emerged, both in terms of chemical composition and in vitro parameters, whereas CNT contained more lipids (EE: 26.4% DM), which, despite being an important energy source, limit microbial activity resulting in low total digestibility, SCFA and gas production. All the parameters related to cashew apple and testa by-products were lower compared to the CM, which were not producing gas after 48 h. However, the differences might be due to the chemical composition, in particular, ADL and EE content and secondary compounds in cashew apple by-products. Previous studies reported a higher content of secondary compounds, principally tannin, in cashew whole apple and pomace [4,10] and cashew nut testa [37,38]. Some in vivo studies on the use of cashew apple in pig diets reported encouraging data. In the Philippines, fresh (20%) or sun-dried (20%) cashew wastes were used as a supplement to a mixed ration of rice bran, corn meal and fish meal in fattening pigs. Animal growth performances were not significantly different from those obtained with the control diet, while feed costs were significantly reduced [39]. In Brazil, dried cashew pomace fed to growing pigs of up to 20% of the diet, compared to soybean meal or sorghum, showed lower protein and energy digestibility (12 and 23%, respectively), thus also reducing the metabolizable energy of the diet from 4125 to 3225 kcal/kg [40]. In addition, it was proposed to soak CNT in distilled water (1:10, w/v) for 24h to reduce tannin and to include it in finishing the pig diet of up to 5% to maintain intake and nutrient digestibility [38]. Overall, cashew by-products (especially pomace and whole fruit) can be an ingredient for pigs due to their protein content, not too high structural carbohydrate content, high digestibility and BCFA production. Importantly, due to their high-water content, it is necessary to find an adequate system to preserve them.

### 4.3. Use of Mango By-Products in Pig Nutrition

The three mango by-products showed different characteristics: (i) the peel had a limited protein content and a low level of non-lignified structural carbohydrates, which led to a high digestibility, high gas and short-chain fatty acids production after 72 h of fermentation; (ii) the testa had a low protein level and a fair content of lignified structural carbohydrates, which contributed to the low digestibility after 72 h, accompanied by an equally low production of gas (slow kinetics) and short-chain fatty acids; and (iii) the kernel showed the best nutritional value with a higher protein content, limited content of structural carbohydrates and a high level of crude lipids, which led to a high degradability, production of gas (faster kinetics) and short-chain fatty acids, in particular butyric acid. This last aspect is particularly interesting in monogastrics because this acid represents an energy source for the enterocyte. In the literature, most of the studies on chemical composition refer to kernel and peel and the data reported are similar to those obtained in the present study for the kernel but slightly different for the peel, especially regarding NDF and CP [41]. An in vivo study reported that dried mango peels can be included at 10% in the diets for finishing pigs without any negative effects on the feed conversion index or animal performance and provide economic advantages [42]. Overall, mango by-products (especially peel and kernel) could represent a raw material in pig feeding due to their high digestibility and high lipid content, but need to be integrated with crude protein sources. However, Timbilfou et al. [14] published that, in the following order according to an economic point of view, palm kernel meal, rice bran, maize bran and wheat bran can be used at 25% as sorbent for mango peel (75%) to produce feed suitable for monogastric and ruminant as well.

## 5. Conclusions

Due to their chemical composition, digestibility and in vitro fermentation characteristics, pineapple pomace, cashew pomace and whole fruit could be included in a diet for fattening pigs, presenting characteristics partly similar to beet pulp. Overall, also mango by-products (especially peel and kernel) could represent a raw material in pig feed, due to the high digestibility and lipid content, even if they have to be integrated with a crude protein source.

In addition, as these by-products are available in tropical and subtropical areas, their recycling in pig nutrition may reduce feeding costs, protect the environment from their decomposition and increase breeders’ income. For all by-products, further studies are needed to find a solution for their preservation (i.e., drying, ensiling). The combination of these by-products with others, which have higher DM as sorbent, should be an option to preserve cashew apple, pineapple and mango by-production for pig nutrition. Furthermore, in vivo tests should assess the appropriate level of integration in a pig diet that does not negatively influence palatability or ingestion and is adequate for growth performance. Nowadays, the tendency is to consider the by-product as a co-product, i.e., a valuable material generated during a production cycle together with others that can be sold or reused profitably. The non-leader co-product must be treated with the same care and attention as the main co-product. Consequently, by-products become a resource, with added socio-economic value if nutritional integrity is preserved. The supply chain that generates them, therefore, should be designed to support it by ensuring not only legal compliance with safety requirements but also the adaptation of processing techniques to improve nutritional quality.

## Figures and Tables

**Figure 1 animals-15-01454-f001:**
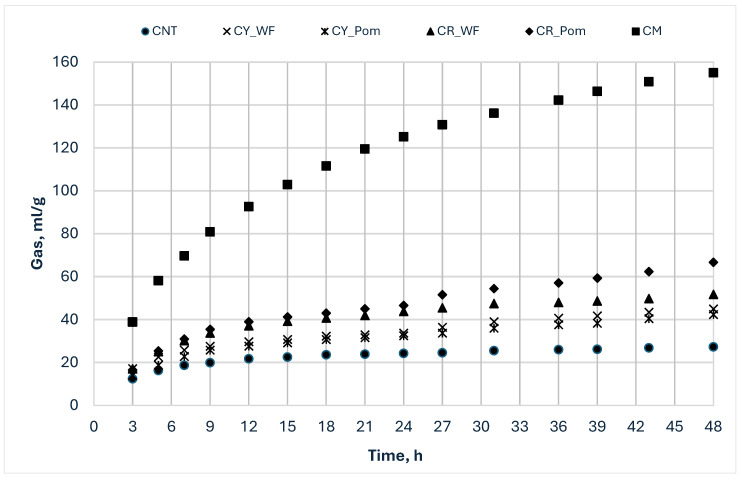
In vitro gas production profiles of by-products incubated for 48 h. CNT: cashew nut testa; CY_WF: cashew (var. yellow) whole fruit; CY_Pom: cashew (var. yellow) pomace; CR_WF: cashew (var. red) whole fruit; CR_Pom: cashew (var. red) pomace; CM: corn meal.

**Figure 2 animals-15-01454-f002:**
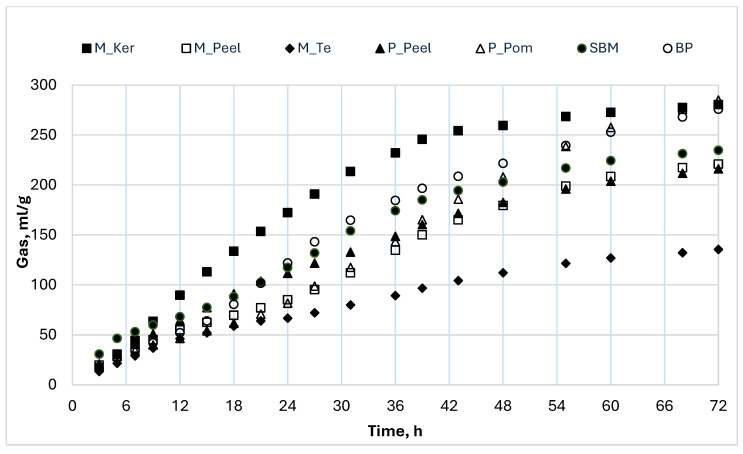
In vitro gas production profiles of by-products incubated for 72 h. P_Peel: pineapple peel; P_Pom: pineapple pomace; M_Peel: mango peel; M_Ker: mango kernel; M_T: mango testa; SBM: soybean meal; BP: beet pulp.

**Table 1 animals-15-01454-t001:** Chemical composition of cashew, pineapple and mango by-products (No. = 3).

Items	DM	Ash	NDF	ADF	ADL	EE	CP	NSC
	%	% DM						
P_Peel	14.51	5.95	35.02	17.22	1.83	0.47	7.77	44.31
P_Pom	17.63	3.91	20.32	9.82	1.70	0.27	9.18	50.20
CNT	91.42	2.25	22.51	17.60	6.79	26.42	11.52	35.63
CY_WF	13.00	3.00	16.13	15.90	7.67	2.19	7.54	61.32
CY_Pom	21.62	2.33	33.82	28.51	11.63	4.73	10.21	43.24
CR_WF	13.35	3.54	19.75	18.42	9.69	1.51	6.21	59.61
CR_Pom	27.22	2.16	33.53	31.80	11.50	3.17	7.73	47.60
M_Peel	22.27	4.45	12.43	11.32	1.52	1.46	2.51	75.54
M_Ker	42.12	2.40	10.90	4.18	0.14	11.23	4.69	68.70
M_T	52.04	3.82	25.32	24.13	8.68	3.66	3.97	59.93
CM	86.71	1.48	7.66	2.83	2.40	1.72	7.65	68.25
SBM	89.32	7.77	14.91	5.07	0.40	0.99	51.23	14.66
BP	24.34	1.50	47.15	23.83	2.70	0.83	9.32	33.02

P_Peel: pineapple peel; P_Pom: pineapple pomace; CNT: cashew nut testa; CY_WF: cashew (var. yellow) whole fruit; CY_Pom: cashew (var. yellow) pomace; CR_WF: cashew (var. red) whole fruit; CR_Pom: cashew (var. red) pomace; M_Peel: mango peel; M_Ker: mango kernel; M_T: mango testa. CM: corn meal; SBM: soybean meal; BP: beet pulp. DM: dry matter; NDF: neutral detergent fiber; ADF: acid detergent fiber; ADL: lignin detergent fiber; EE: ether extract; CP: crude protein; NSC: non-structural carbohydrates.

**Table 2 animals-15-01454-t002:** In vitro two-step DM digestibility (%) of cashew, pineapple and mango by-products.

Items	DMDe	DMDm	DMDt
Incubation at 48 h			
CNT	18.8 ^e^	12.0 ^d^	28.5 ^e^
CY_WF	60.0 ^b^	24.3 ^b^	69.7 ^b^
CY_Pom	30.0 ^d^	24.7 ^b^	47.3 ^c^
CR_WF	61.4 ^b^	20.4 ^c^	69.3 ^b^
CR_Pom	38.0 ^c^	10.6 ^d^	44.6 ^d^
CM	67.3 ^a^	64.1 ^a^	88.3 ^a^
*p*-value	<0.0001	<0.0001	<0.0001
MSE	2.742	0.530	0.837
Incubation at 72 h			
P_Peel	50.3 ^b^	59.0 ^e^	79.6 ^d^
P_Pom	68.3 ^a^	68.8 ^cd^	90.1 ^b^
M_Peel	48.4 ^b^	70.1 ^c^	84.6 ^c^
M_Ker	10.0 ^c^	67.4 ^d^	70.7 ^e^
M_T	11.8 ^c^	24.7 ^f^	33.6 ^f^
SBM	65.2 ^a^	83.2 ^a^	94.2 ^a^
BP	64.0 ^a^	78.0 ^b^	92.1 ^b^
*p*-value	<0.0001	<0.0001	<0.0001
MSE	8.07	0.907	0.752

P_Peel: pineapple peel; P_Pom: pineapple pomace; CNT: cashew nut testa; CY_WF: cashew (var. yellow) whole fruit; CY_Pom: cashew (var. yellow) pomace; CR_WF: cashew (var. red) whole fruit; CR_Pom: cashew (var. red) pomace; M_Peel: mango peel; M_Ker: mango kernel; M_T: mango testa; CM: corn meal; SBM: soybean meal; BP: beet pulp. DMDe: enzymatic digestibility (No. = 4); DMDm: microbial digestibility (No. = 3); DMDt: total digestibility. MSE: mean square error. Along the column, different letters indicate statistically significant differences *p* < 0.05.

**Table 3 animals-15-01454-t003:** In vitro fermentation characteristics after 48 h and 72 h of cashew, pineapple and mango by-products (No. = 6).

Items	pH	OMCV	Ace	Pro	Iso-But	But	Iso-Val	Val	SCFA	BCFA
		ml/g	mmol/g							
	Incubation at 48 h									
CNT	6.25 ^AB^	27.31 ^D^	11.62 ^D^	3.04 ^E^	0.22 ^D^	2.55 ^B^	0.54 ^D^	0.51 ^D^	18.52 ^D^	4.12 ^D^
CY_WF	6.25 ^AB^	45.05 ^C^	20.44 ^B^	7.23 ^BC^	1.34 ^B^	3.85 ^B^	2.50 ^A^	1.04 ^AC^	36.44 ^BC^	10.63 ^AB^
CY_Pom	6.24 ^B^	42.03 ^C^	17.43 ^C^	6.20 ^CD^	0.97 ^C^	3.40 ^B^	2.11 ^A^	1.74 ^B^	31.73 ^BC^	9.66 ^B^
CR_WF	6.32 ^A^	52.06 ^BC^	17.44 ^C^	5.34 ^D^	1.23 ^B^	3.09 ^B^	2.25 ^A^	1.28 ^C^	30.60 ^C^	11.45 ^A^
CR_Pom	6.20 ^B^	58.05 ^B^	20.72 ^B^	7.54 ^B^	0.96 ^C^	3.91 ^B^	1.54	1.96 ^B^	36.73 ^B^	6.84 ^C^
CM	5.75 ^C^	142 ^A^	38.51 ^A^	13.13 ^A^	1.90 ^C^	26.0 ^A^	2.47 ^A^	5.86	87.92 ^A^	4.98 ^D^
*p*-value	<0.0001	<0.0001	<0.0001	<0.0001	<0.0001	<0.0001	<0.0001	<0.0001	<0.0001	<0.0001
MSE	0.003	28.52	2.569	0.387	0.015	1.273	0.05	0.033	11.02	0.467
	Incubation at 72 h									
P_Peel	5.90 ^B^	226 ^C^	59.42 ^AB^	17.81 ^BC^	1.57	15.02 ^B^	2.20	2.58	98.52	3.84 ^C^
P_Pom	5.63 ^A^	285 ^A^	64.81 ^A^	26.74 ^A^	1.73	12.45 ^B^	2.34	3.02	111.00	2.67 ^D^
M_Peel	5.75 ^C^	221 ^C^	55.10 ^A^	20.13 ^BC^	1.65	9.02 ^B^	2.33	2.39	90.64	4.38 ^B^
M_Ker	5.70 ^CD^	281 ^A^	43.74 ^CD^	26.66 ^A^	1.58	25.72 ^A^	2.26	2.13	102.01	3.73 ^C^
M_T	6.12 ^A^	136 ^D^	32.02 ^D^	12.84 ^C^	1.45	12.75 ^AB^	2.89	2.99	69.92	6.23 ^A^
SBM	5.84B ^C^	253 ^B^	49.63 ^BC^	28.14 ^A^	1.87	13.48 ^B^	2.36	3.09	98.43	4.30 ^B^
BP	5.61 ^D^	276 ^A^	60.30 ^AB^	24.22 ^AB^	1.43	16.83 ^AB^	2.10	2.86	108.14	3.29 ^CD^
*p*-value	<0.0001	<0.0001	<0.0001	<0.0001	<0.0001	<0.0001	<0.0001	<0.0001	0.147	<0.0001
MSE	0.007	47.37	46.19	22.56	0.100	33.10	0.223	0.143	290	0.061

P_Peel: pineapple peel; P_Pom: pineapple pomace; CNT: cashew nut testa; CY_WF: cashew (var. yellow) whole fruit; CY_Pom: cashew (var. yellow) pomace; CR_WF: cashew (var. red) whole fruit; CR_Pom: cashew (var. red) pomace; M_Peel: mango peel; M_Ker: mango kernel; M_T: mango testa. CM: corn meal; SBM: soybean meal; BP: beet pulp. OMCV: cumulative volume of gas related to incubated OM; SCFA: short-chain fatty acids; Ace: acetate; Pro: propionate; Iso-But: iso-butyrate; But: butyrate; Iso-Val: iso-valerate; Val: valerate; BCFA: branched-chain fatty acids. MSE: mean square error. Along the column, different letters indicate statistically significant differences *p* < 0.05.

## Data Availability

The data that support this study will be shared upon reasonable request to the corresponding author.

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
