# Peer review of "Potential Use of Tropical and Subtropical Fruits By-Products in Pig Diet: In Vitro Two-Step Evaluation"

_animals, 2025, doi:10.3390/ani15101454_

Round 1

Reviewer 1 Report

Comments and Suggestions for Authors

The following issues need to be revised:

  1. In the article, the digestion characteristics of unconventional feed resourceswere determined by the in vitro two-step method. Please verify whether the cited references are correct, and it is recommended to refer the methods from the latest research literature on pigs;
  2. The shaking flask method was used, and the enzymatic hydrolysis substrate will inhibit the enzymatic hydrolysis digestion process,whether the measured result was low? If other literatures do not adopt the similar methods, it will also be difficult to compare the differences in values among the literatures in discussion section.
  3. Please add the number of samples measured "N=?" in the notes of Table 1-Table 3;
  4. These types of raw materials have a high moisture content and are not easy to preserve. Have you considered the feed hygiene indicators, especially the content of mycotoxins? If measured, it is recommended to supplement or describe the measurement results.
  5. There are some minor errors in the article that need to be corrected, such as Line 72 106, Line174 CO2.

Author Response

Please find attached the replies to reviewer comments.

Reviewer 2 Report

Comments and Suggestions for Authors

The present work is an in vitro investigation of the potential use of Tropical and Subtropical Fruits by-products (Pineapple, cashew, and mango) in pig nutrition. This study has the merit of highlighting the chemical composition, the in vitro digestibility, fermentation, and gas production of these by-products as feed ingredients for swine diets. In addition, this article also discusses how pineapple, cashew, and mango by-products could be practically incorporated in pig diets to take part in reducing the worldwide amount of food waste. 

L48: The references (1 and 2) mentioned in this line are not present in the reference list.

L53-L54: "In 2021, global pineapple ...tons": add a reference to this information

L99-L107: I think the title of this part should not be experimental design, but maybe "Materials and locations of the study". Anyways, you should rename it.

L144-186: The title of this part should be: "Experimental design and in vitro two-step digestibility evaluation"

Comments on the Quality of English Language

The overall article is readable and understandable. However, a few parts need to be addressed. 

L19-L21: "From harvesting... its pollution": This sentence needs to be rewritten

L36: "In general" instead of "in general"

L37: Please write "such as" instead of "eg"

L38: Please, delete "mentioned"

L77: Please write "including" instead of "eg"

L79: Please, delete "mentioned"

L214: Please delete "where"

L264: Please write "produced" instead of "prodzuced"

L313: "lies in" instead of "lies to"

L359: Delete "in" in this line.

L371: Please, write "up to 5%"

L390: Write "An in vivo" instead of "In In vivo"

Author Response

(The authors gave the same response as above.)

Reviewer 3 Report

Comments and Suggestions for Authors

In this study, Dieu donné Kiatti et al., aimed to evaluate through an in vitro two-steps (gastro-intestinal and caecum) study, ten tropical and subtropical fruits by-products derived from the processing of pineapple, cashew and mango for pig nutrition to provide feeds not competing with humans and promoting eco-sustainable livestock. It’s an interesting study, however, some modifications and clarifications are required:

Abstract:

Line 30: "...... and protein (7.65 – 51.2% DM) ......". In table 1 the interval is "...... and protein (6.21 – 51.2% DM) ......". Please check the data of the Table 1.

Introduction:

Clearly establish the context for the study.

There are some inconsistencies in the bibliography (for example: lines 91, 92, and so on). Please check and correct number of the references.

Materials and Methods:

The study design is interesting; however, it needs some improvements.

There are some inconsistencies in the bibliography (for example: lines 147, 149, and so on). Please check and correct number of the references.

Experimental design

Line 103: You mentioned that "All the samples were oven dried at 65°C ……", without mentioning how many hours were they dried. Please specify.

Samples collection

Line 135: Please correct this: "… were peel, kernel, taste …" with "… were peel, kernel, testa …".

In vitro two-step digestibility evaluation

Line 164: You mentioned that "… substrate were dried overnight at 65°C …", without mentioning how many hours were they dried. Please specify.

Lines 171 to 172: You mentioned that "... collected in heated thermoses …", without mentioning at what temperature were they heated. Please specify.

Line 174: You mentioned that "... already containing the medium (79 ml) ……", without mentioning what the medium contained. Please specify.

Line 175: You mentioned that "... were incubated at 39°C", without mentioning under what conditions were they incubated. Please clarify.

Lines 175 to 177: You mentioned that "Depending on the chemical composition, the incubation was stopped at different times: i) at 48 hours: CNT, CY_WF CY_Pom, CR_WF, CR_Pom and CM; ii) at 72 hours: P_Peel, P_Pom, M_Peel, M_Ker, M_Te, SBM and BP." Please specify exactly what were the criteria for which the incubation was stopped at 48 and 72 hours, respectively.

Lines 188 to 195: I recommend that you give a reference for the equations (1), (2) and (3).

Results

The results are clear and concise; however, it needs some corrections.

Please use the decimals consistently in the tables 1-3.

Line 205: Please correct this: "… CNT, CM and SBM (91.4, 89.3 and 86.9 %, respectively) …" with "… CNT, CM and SBM (91.4, 86.7 and 89.3 %, respectively) …". Please check the data of the Table 1.

Lines 205 to 206: You mentioned that "....... were low (24.8 ± 13.7 %)". Please check the data of the Table 1.

Lines 210 to 211: You mentioned that "In all the other feedstuffs NDF values fall between 20 and 10 % DM, ....". Please check the data of the Table 1.

Line 215: You mentioned that "....... higher (54.3 ± 13.1 % DM)". Please check the data of the Table 1.

Line 256: You mentioned that "....... considered physiological in pig cecum-colon (5.2 – 7.8)". I recommend that you give a reference.

Line 264: Please correct this: "..., the substrates which prodzuced …" with "..., the substrates which produced …".

Line 265: You mentioned that "....... and the one which produced the least was P_Peel …". Please check the data of the Table 3.

Line 268: You mentioned that "....... were significantly higher in M_Te and lower in BP (6.23 and 3.29; …". Please check the data of the Table 3.

Lines 289 to 290: In the table 3, you mentioned value of BCFA in P_Pom to be 2.67 mmol/g at 72 hours of incubation. Please check and correct.

Discussion

Well done and sustained the results; however, it needs some corrections.

Line 317: You mentioned that "....... and CP (7.65 – 51.2% DM) except …". Please check the data of the Table 1.

Lines 360 to 362: You mentioned that "Previous study reported higher content ... cashew nut testa [36, 37]". Please check and correct number of the reference.

Lines 387 to 390: You mentioned that "In the literature, most of the studies on the chemical composition refer to kernel and peel ....... but slightly different in the peel especially regarding NDF and CP [40]". Please check and correct number of the reference.

Lines 390 to 392: You mentioned that "In in vivo studies reported that dried mango peels can be included …….. and providing economical advantageous [41]". Please check and correct number of the reference.

Line 395: You mentioned that " However, Timbilfou et al. [14] published that palm kernel meal, …". Please check and correct number of the reference, respectively [16].

Conclusions

The conclusions are clear and reflect well the results obtained in the your research.

References

There are some inconsistencies in the bibliography (for example: lines 91, 92, 147, 149, 362, 390, 392, 395, and so on). Please check all citations in the manuscript and correct number of the references.

Author Response

(The authors gave the same response as above.)
